

# The community composition variation of Russulaceae associated with the *Quercus mongolica* forest during the growing season at Wudalianchi City, China

Pengjie Xing[1,*], Yang Xu[1,*], Tingting Gao[1], Guanlin Li[1], Jijiang Zhou[1], Mengle Xie[1,2] and Ruiqing Ji[1]

[1] Engineering Research Center of Edible and Medicinal Fungi, Jilin Agricultural University, Changchun, China
[2] Life Science College, Northeast Normal University, Changchun, China
* These authors contributed equally to this work.

Corresponding author
Ruiqing Ji, jiruiqingjrq@126.com

## ABSTRACT

**Background:** Most species of the Russulaceae are ectomycorrhizal (ECM) fungi, which are widely distributed in different types of forest ecology and drive important ecological and economic functions. Little is known about the composition variation of the Russulaceae fungal community aboveground and in the root and soil during the growing season (June–October) from a *Quercus mongolica* forest. In this study, we investigated the changes in the composition of the Russulaceae during the growing season of this type of forest in Wudalianchi City, China.

**Methods:** To achieve this, the Sanger sequencing method was used to identify the Russulaceae aboveground, and the high-throughput sequencing method was used to analyze the species composition of the Russulaceae in the root and soil. Moreover, we used the Pearson correlation analysis, the redundancy analysis and the multivariate linear regression analysis to analyze which factors significantly affected the composition and distribution of the Russulaceae fungal community.

**Results:** A total of 56 species of Russulaceae were detected in the *Q. mongolica* forest, which included 48 species of *Russula*, seven species of *Lactarius*, and one species of *Lactifluus*. *Russula* was the dominant group. During the growing season, the sporocarps of *Russula* appeared earlier than those of *Lactarius*. The number of species aboveground exhibited a decrease after the increase and were significantly affected by the average monthly air temperature ($r = -0.822$, $p = 0.045$), average monthly relative humidity ($r = -0.826$, $p = 0.043$), monthly rainfall ($r = 0.850$, $p = 0.032$), soil moisture ($r = 0.841$, $p = 0.036$) and soil organic matter ($r = 0.911$, $p = 0.012$). In the roots and soils under the *Q. mongolica* forest, the number of species did not show an apparent trend. The number of species from the roots was the largest in September and the lowest in August, while those from the soils were the largest in October and the lowest in June. Both were significantly affected by the average monthly air temperature ($r^2 = 0.6083$, $p = 0.040$) and monthly rainfall ($r^2 = 0.6354$, $p = 0.039$). Moreover, the relative abundance of *Russula* and *Lactarius* in the roots and soils showed a linear correlation with the relative abundance of the other fungal genera.

## INTRODUCTION

Almost all the land plant species form a mutualistic symbiosis with mycorrhizal fungi, including endo- and ectomycorrhizal (ECM) fungi. ECM fungi can favor the nutrient uptake of the host, promote plant growth and enhance plant drought resistance and disease resistance (*Bryla & Duniway, 1997*; *Brundrett & Kendrick, 1988*; *Harley & Smith, 2008*; *Smith & Read, 2008*; *Courty et al., 2010*; *Dickie et al., 2015*; *Lofgren, Nguyen & Kennedy, 2018*; *Pacé et al., 2019*), and at the same time, complete their life histories through the absorption of essential nutrients from the soil and the roots of the host (*Karwa, Varma & Rai, 2011*). On a global scale, the ECM fungi have a rich diversity and an extensive distribution. These ECM fungi play an essential role for host trees and contribute to the proper functioning of forest ecosystems. The composition of the ECM fungal community has received much attention from researchers.

The community composition of ECM fungi in forest ecosystems shows monthly changes. Previous research showed that ECM community species richness in the *Quercus ilex* L. roots was the highest in the autumn and the lowest in the summer (*Román & Miguel, 2005*). The ECM root tip abundance changed with the season variation (*Blasius, Kottke & Oberwinkler, 1990*). The peak productivity of the Russulaceae fungi in the *C. cuspidata* forest was in mid-summer (*Murakami, 1987*). Moreover, the composition and distribution of the ECM fungal community have been influenced by biotic factors and abiotic factors.

In biotic factors, the ECM fungi not only interact with their host (*Kernaghan et al., 2003*; *Izzo, Agbowo & Bruns, 2005*; *Vasco-Palacios et al., 2018*) but also with other soil fungi, including other ECM, saprotrophic and pathogenic fungi (*Cairney & Meharg, 2002*; *Kennedy, 2010*; *Mundra et al., 2016*). Abiotic factors in the climate and soil have strong effects on the ECM fungal community. Warming could increase the abundance of the ECM fungi in the roots (*Deslippe et al., 2011*; *Saravesi et al., 2019*). Abundant rainfall and soil moisture are necessary for the fungi to fruit. If the SM potential was too low, the fungi were unable to obtain sufficient water for fruit body development (*Salerni et al., 2002*; *Barroetaveña, La Manna & Alonso, 2008*). The soil pH was also the primary factor affecting the evenness of the belowground mycorrhizal communities (*Suz et al., 2014*). Additionally, the complementarity of P uptake was affected by the diversity of the ECM fungal species, and P uptake efficiency is related to soil moisture; the efficiency of the P uptake decreased when the soil moisture was limiting (*Köhler et al., 2018*). Long-term N deposition in the soil can lead to a decline in the richness in the ECM fungal species and dramatic changes in the ECM fungal community structure (*Lilleskov et al., 2002*; *Hedwall et al., 2018*).

Most of the research on the ECM fungal community microecology described above was based solely on samples from aboveground or belowground (*Villarreal-Ruiz & Neri-Luna, 2018*). However, there are some contradictions on the ECM community between the

aboveground and belowground, while some common fruiting species produced few mycorrhizae; some common species observed on the roots were poorly represented or entirely lacking in the aboveground fruiting record (*Gardes & Bruns, 1996*). Therefore, it is necessary to study the ECM fungal community combining the aboveground and belowground species (*Dickie et al., 2009*; *Wei et al., 2018*).

Almost all of the species of the Russulaceae are ECM fungi and are widely distributed in temperate regions. They can be associated with species of pine, oak, fir and spruce (*Bills, Holtzman & Miller, 1986*; *Kernaghan, Currah & Bayer, 1997*; *Adamčík, Jančovičová & Valachovič, 2013*; *Wang et al., 2015*; *Lazarević & Menkis, 2018*; *Wang et al., 2017*).

*Quercus mongolica* Fisch. ex Ledeb. is one of the quickest growing, high-yield plantation tree species and comprises the primary commercial forests in Northeast China. The symbiotic relationship between the Russulaceae fungi and *Q. mongolica* plays an important role in maintaining the regional ecological balance and enabling ecosystem restoration and reconstruction. Currently, most researchers have focused on the composition of the Russulaceae community in oak forests. The Russulaceae fungi are dominant in the oak forests of North America (*Dickie et al., 2009*) and the *Q. mongolica* forest of Inner Mongolia China (*Wei et al., 2018*) from both the aboveground and belowground. However, the components of the microecology of the Russulaceae fungal community in the *Q. mongolica* forest are unclear and merit further study.

In our study, based on the investigation of the Russulaceae fungal community from aboveground and belowground in different months during the growing season from June to October, we aimed to study: (1) what the trends of the changes in the community composition of the Russulaceae fungi are during the growth season in the *Q. mongolica* forest, and (2) how abiotic factors and biotic factors affect the community composition of the Russulaceae fungi. We hypothesized that (1) during the growing season, the species number of Russulaceae fungi aboveground would increase first and then decrease; the species number of Russulaceae fungi in roots would remain basically unchanged, and the species number of Russulaceae fungi in soils would increase; (2) temperature and soil pH were important factors affecting the composition and distribution of the Russulaceae fungi aboveground and belowground; (3) fungi sporocarp diversity (including Russulaceae) on the above-ground and in the root system mostly was influenced by the soil fungal community.

## MATERIALS AND METHOD

### Study site

Our study site, the *Q. mongolica* forest, is located at the Jiaodebu forest farm construction area, Wudalianchi Scenic Area nature reserve, Heilongjiang Province, China (126° 11′ 0 E, 48° 37′ 0 N, 303 m asl) (our researches has been allowed), which is almost covered by volcanic ash soil, and is a temperate continental monsoon climate with an average temperature of 0 °C. The area receives approximately 600–900 mm rainfall per year with peaks during June–August.
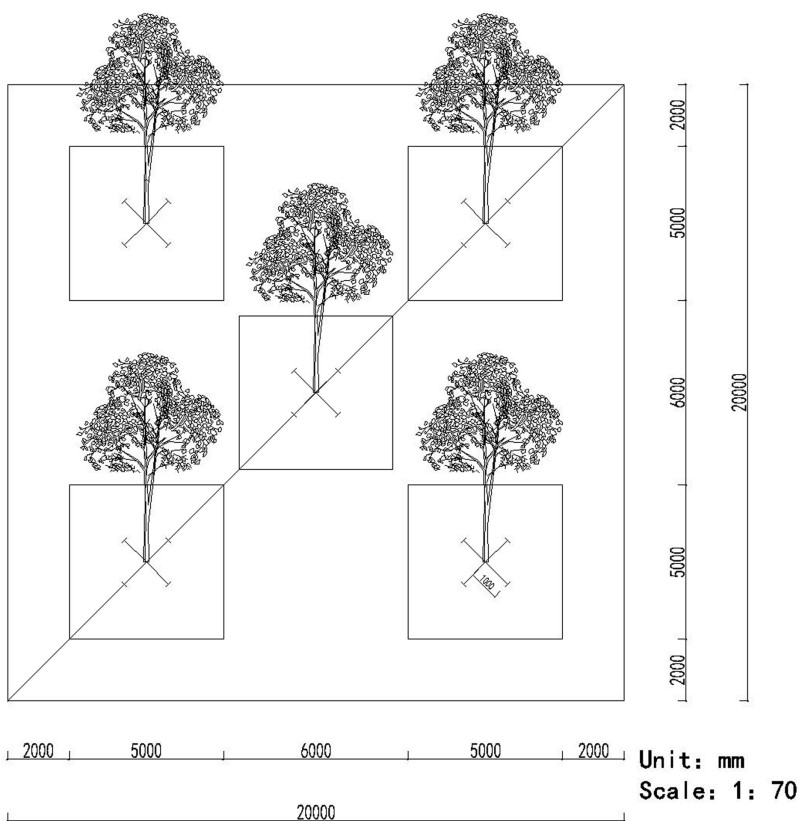

**Figure 1 The pattern of the five-point method for sampling the roots and soils.**

## Sampling strategy

Four 20 m × 20 m plots were surveyed, with an interval of over 200 m between plots at the site, which is composed of *Q. mongolica* (98%) and *Betula platyphylla* Sukaczev (2%). We collected samples twice per month from June 2018 to October 2018.

The sporocarps of the Russulaceae fungi were randomly acquired and fully counted in each plot. Sporocarps were photographed in the field using a Canon EOS 70D digital camera. Fresh morphological characters were recorded, and the colors were designated using the Munsell Color System (*Jabeen, Niazi & Khalid, 2016*). Some sporocarps of each species were selected and contained in zip lock bags with silica gel, while the others were dried by an oven (approximately 50 °C) and placed into specimen boxes.

The root samples were collected using a five-point sampling method, that is, five sampling points were distributed in the middle and four corners of the plot, respectively. In each point, we randomly selected a target tree, collected the fine root segments (approximately 15–30 cm long) in four directions of 1 m from the trunk of the tree (*Wang et al., 2017*) (Fig. 1). Simultaneously, a soil sample was also collected using the five-point sampling method (Fig. 1). Eight cylinders of soil were collected randomly from each point (approximately 12 cm deep and 5 cm in diameter) to remove the impurities and then mixed (more than 500 g) together (*Castaño et al., 2016*). All the root and soil samples from

the five points during the same month were homogenized, respectively, and pooled to obtain a composite sample, which was placed into a cooler containing ice and transported to our laboratory.

## Sample processing

We identified the Russulaceae fungal sporocarps based on morphological observation and molecular identification methods. Anatomical features were measured using a Zeiss Lab. A1 microscope. For detailed anatomical examination, tissues from the specimens were mounted on glass slides and observed in Phloxine (1%) for better contrast and Melzer's KOH (5%) for colored hyphae. We measured basidia ($n = 10$/sample) and basidiospores ($n = 20$/sample) as length range × width range (*Jabeen, Niazi & Khalid, 2016*).

The DNA of the sporocarps was extracted using a CTAB method (*Gardes & Bruns, 1993*). A polymerase chain reaction (PCR) with the primer pairs ITS-1F and ITS-4 or ITS4-B (*White et al., 1990*; *Gardes & Bruns, 1993*) was conducted, and finally the PCR products were sequenced using the Sanger method. The sequences we obtained were BLASTed against the NCBI database and UNITE database at a 98% sequence identity threshold, and undescribed species were identified to the genus. Sequences generated in this study were submitted to GenBank.

Each root sample was soaked in Tween 20 (0.1% in water) for approximately 1 h and was washed well under running water to remove adhering soil particles. They were then dried completely by placement in a container filled with silica beads. The dried roots were placed in new bags and crumbled to break off the fine roots of the higher order roots. The powdered fine root material were placed into a sterile 50 ml centrifuge tube for DNA extraction (*Benucci & Bonito, 2016*) and high-throughput sequencing.

DNA was extracted from the root sample using a Soil Isolation Kit (Macherey–Nagel, Duren, Germany). The ITS1 region was amplified with primers (ITS1: GTTGGTCATTT AGAGGAAGTAA; ITS2: GCTGCGTTCTTCATCGATGC) using the manufacturer's instructions. The PCR was conducted in 50 μl reactions consisting of 30 ng genomic DNA, fourμl PCR Primers, 25 μl PCR Master Mix, and ddH$_2$O as needed. The PCR reactions were run under the following conditions: 98 °C for 3 min, followed by thirty cycles of 98 °C for 45 s, 55 °C for 45 s, 72 °C for 45 s, and a final extension step at 72 °C for 7 min before storage at 4 °C. The PCR products were purified with Ampure XPbeads (Agencourt Biosciences, Beverly, MA, USA) to remove the unspecific products. The samples were then pooled in equimolar concentrations for paired-end sequencing on an Illumina Hi-Seq 2500 platform in BGI Co., Ltd, Beijing, China.

The aluminum box with 20.0 g of fresh soil obtained with a 2 mm sieve in an oven at 105 °C was dried to a constant weight to measure the soil moisture (SM). Some soil was completely dried by placement in a container filled with silica beads for days. The dried soil stone particles that had been removed with the 2 mm sieve were used for measuring other soil parameters. The soil available phosphorus (P) was determined using 0.5 mol·L$^{-1}$ of the NaHCO$_3$ extraction-molybdenum anti-colorimetric method. The soil organic matter (SOM) was detected using the potassium dichromate volumetric method. The soil effective nitrogen (N) was measured by the alkali diffusion method.

The soil available potassium (K) was determined using 1 mol·L$^{-1}$ NH$_4$OAc extraction-flame photometry (*Bao, 2000*). The soil pH was measured by potentiometry.

Some fresh soil samples were placed into a sterile 50 ml centrifuge tube for DNA extraction and high-throughput sequencing. The DNA extraction methods, PCR and sequencing conditions were conducted with the root samples in the same manner as the soil analyses.

## Bioinformatics

The raw data were filtered to eliminate the adapter pollution and low quality to obtain clean reads. After that, the paired-end reads with overlap were merged to tags. The tags were clustered to operational taxonomic units (OTUs) using USEARCH (v 7.0.1090) (*Edgar, 2013*) at 97% sequence similarity. OTU representative sequences were taxonomically classified using Ribosomal Database Project RDP Classifier (v. 2.2) trained on the database UNITE (v. 6) using 0.5 confidence values. At last, the alpha diversity was analyzed based on the OTU and taxonomic ranks. The high-throughput sequencing raw data of fungus in root and soil samples were uploaded into Sequence Read Archive (SRA) in NCBI (SRA accession numbers: SRR10590054–SRR10590058, SRR10590043–SRR10590047).

## Statistical analyses

Three alpha diversity indices were used to analyze the community composition of the Russulaceae fungi aboveground. The Menhinick richness index (D1) reflected the species richness of the community. The Shannon index (D2) reflected the diversity of the community species. The Pielou's evenness index (D3) reflected the evenness of the distribution of the number of individuals in each species. The diversity index formulas were as follows:

- D1 = S/√(N), S is the total number of species in the community; $N$ is the total number of individuals observed;
- D2 = $-\sum P_i \text{Ln}(P_i)$, $P_i$ is the proportion of individuals found belonging to the $i$th species; ln is the natural logarithm;
- D3 = H′/H′ max, H′ max = Ln(S), H′ = D2.

The alpha diversity index (i.e., the Chao and Shannon indices) in the root and soil samples were calculated using Mothur (v 1.31.2). A one-way analysis of variance (ANOVA) was performed on the alpha diversity index difference of the sporocarps aboveground and the Russulaceae fungi in the root and soil from different months, respectively, and was conducted to analyze the difference of the Shannon index between the aboveground and roots and between the aboveground and soil, respectively, in July and August, with Tukey's HSD test at $p < 0.05$ in SPSS (v 19.0). The Wilcoxon Rank–Sum test was used to compare the differences in the alpha diversity index between the root and soil groups, and then a plotbox of the alpha diversity drawn by R. Linear Discriminant Analysis (LDA) was used to examine the significant difference in the Russulaceae

species via Lefse software. A Venn diagram was used to visualize the shared number of Russulaceae species between above- and belowground.

Abiotic factors were stated and included monthly rainfall (MR), average monthly relative humidity (RH), average month air temperature (Temp), SM, SOM, P, N, K and pH. The data for the MR, RH and Temp were obtained from the Dazhanhe Meteorological Station in Wudalianchi City, Heilongjiang Province, China.

The Pearson correlation was utilized to follow the relationship between the alpha diversity index of the Russulaceae sporocarps against these abiotic factors (*p*-values, two-tailed; confidence intervals, 95%) using SPSS software. In addition, the redundancy analysis (RDA) was used to analyze the correlation between the Russulaceae fungal community composition of the belowground and these variables using the R package *Vegan* (v 2.0-10).

The multivariate linear regression analysis in SPSS was used to detect the relative abundance correlation of other genera with *Russula* and *Lactarius* in the root and soil, respectively. We assigned OTUs to the functional groups using the online application FUNGuild ("http://www.stbates.Org/guilds/app.php") (*Nguyen et al., 2016*). Only FUNGuild assignments at the confidence level of "highly probable" and "probable" were used for the analysis.

## RESULTS

### The composition variation on the growing seasonal basis of the Russulaceae fungal community on aboveground

The fruiting period of the Russulaceae sporocarps aboveground took place in July and August. A total of 106 sporocarps of the Russulaceae were collected and were classified as 24 different species by morphology and Sanger sequence methods, which included 19 species of *Russula*, 4 species of *Lactarius* and 1 species of *Lactifluus* (Table 1).

The ANOVA analysis indicated that the Menhinick index was the highest in August, significantly higher than that in June, July, September and October, and there was no significant difference between these months. The Shannon index was the highest in August, significantly higher than the other months, and July was also significantly higher than June, September and October, with no significant difference between these 3 months. Pielou's evenness index in July was significantly higher than that in June, August, September and October, and there was no significant difference between that in June, September and October (Table 2).

The species number of Russulaceae found aboveground during the growing season showed a trend of increasing first and then decreasing. Species of *Russula* appeared from early July, and reached their numerical peak in the middle of August, gradually decreased at the end of August, and there were almost no sporocarps of *Russula* in early September. The species of *Lactarius* appeared at the end of August and reached their numerical peak in the end of August. In September, its numbers quickly decreased. There was only one species of *Lactarius* in September, and no sporocarps in October. The species

**Table 1 The species of Russulaceae from above-ground, root and soil in different months.**

| Species name | Above-ground | | | | | Root | | | | | Soil | | | | |
|---|---|---|---|---|---|---|---|---|---|---|---|---|---|---|---|
| | June | July | August | September | October | June | July | August | September | October | June | July | August | September | October |
| *Lactarius mammosus* Fr. | – | – | – | – | – | – | – | – | – | √ | – | – | – | – | – |
| *Lactarius pyrogalus* (Bull.) Fr. | – | – | – | – | – | √ | – | – | √ | – | √ | √ | √ | – | √ |
| *Lactarius torminosus* (Schaeff.) Gray | – | – | √ | √ | – | – | √ | – | √ | – | – | – | – | – | – |
| *Lactarius trivialis* (Fr.) Fr. | – | – | √ | – | – | √ | – | – | – | – | √ | – | – | √ | √ |
| *Lactarius vietus* (Fr.) Fr. | – | – | √ | – | – | – | – | √ | – | √ | – | √ | – | – | √ |
| *Lactarius evosmus* Kühner & Romagn. | – | – | √ | – | – | – | – | – | – | – | – | – | – | – | – |
| *Lactarius sp* | – | – | – | – | – | √ | – | – | – | – | – | √ | – | √ | – |
| *Lactifluus bertillonii* (Neuhoff ex Z. Schaef.) Verbeken | – | – | √ | – | – | – | – | √ | – | √ | √ | √ | √ | – | √ |
| *Russula anthracina* Romagn. | – | – | – | – | – | √ | √ | √ | √ | √ | √ | √ | √ | √ | √ |
| *Russula atroglauca* Einhell. | – | – | √ | – | – | – | √ | – | – | – | – | √ | √ | √ | – |
| *Russula aurata* Fr. | – | √ | √ | – | – | – | – | – | – | – | – | – | – | – | – |
| *Russula azurea* Bres. | – | – | – | – | – | √ | √ | √ | √ | √ | √ | √ | √ | √ | √ |
| *Russula odorata* Romagn. | – | – | √ | – | – | √ | – | √ | √ | √ | √ | √ | √ | √ | √ |
| *Russula cremeoavellanea* Singer | – | – | – | – | – | √ | – | √ | – | √ | √ | – | √ | – | √ |
| *Russula cyanoxantha* (Schaeff.) Fr. | – | – | – | – | – | √ | √ | √ | √ | √ | √ | √ | √ | √ | √ |
| *Russula delica* Fr. | – | √ | √ | – | – | – | – | – | – | – | – | – | – | – | – |
| *Russula exalbicans* (Pers.) Melzer & Zvára | – | – | – | – | – | √ | – | – | – | √ | – | – | √ | √ | √ |

| Species name | Above-ground | | | | | Root | | | | | Soil | | | | |
|---|---|---|---|---|---|---|---|---|---|---|---|---|---|---|---|
| | June | July | August | September | October | June | July | August | September | October | June | July | August | September | October |
| *Russula foetens* Pers. | – | – | √ | – | – | √ | √ | √ | – | √ | √ | √ | √ | √ | √ |
| *Russula font-queri* Singer | – | √ | – | – | – | √ | – | – | – | – | – | √ | √ | – | – |
| *Russula globispora* (J. Blum) Bon | – | – | √ | – | – | √ | √ | √ | √ | √ | √ | √ | √ | √ | √ |
| *Russula maculata* Quél. | – | – | – | – | – | √ | – | – | √ | √ | – | √ | – | √ | √ |
| *Russula acrifolia* Romagn. | – | √ | √ | – | – | – | – | – | – | – | – | – | – | – | – |
| *Russula olivobrunnea* Ruots. & Vauras | – | – | – | – | – | – | √ | – | √ | – | – | – | – | √ | √ |
| *Russula pallidospora* J. Blum ex Romagn. | – | – | – | – | – | √ | – | – | √ | √ | √ | √ | √ | √ | √ |
| *Russula pelargonia* Niolle | – | – | – | – | – | – | – | – | – | – | √ | √ | – | – | √ |
| *Russula persicina* Krombh. | – | √ | √ | – | – | √ | √ | √ | √ | √ | √ | √ | √ | √ | √ |
| *Russula puellula* Ebbesen, F.H. Møller & Jul. Schäff. | – | – | – | – | – | √ | – | – | √ | √ | √ | √ | √ | – | √ |
| *Russula romellii* Maire | – | – | – | – | – | – | – | – | – | – | – | √ | – | √ | – |
| *Russula subrubescens* Murrill | – | – | – | – | – | – | √ | √ | √ | – | √ | √ | √ | √ | √ |
| *Russula velenovskyi* Melzer & Zvára | – | √ | – | – | – | √ | – | – | – | – | – | – | – | – | – |
| *Russula vitellina* Gray | – | – | – | – | – | – | √ | – | √ | √ | √ | √ | √ | – | √ |
| *Russula xerampelina* (Schaeff.) Fr. | – | √ | √ | – | – | √ | √ | √ | √ | √ | √ | √ | √ | √ | √ |
| *Russula sp 1* | – | – | – | – | – | √ | √ | √ | √ | √ | √ | √ | √ | √ | √ |
| *Russula sp 2* | – | – | – | – | – | √ | √ | √ | √ | √ | √ | √ | √ | √ | √ |
| *Russula sp 3* | – | – | – | – | – | – | – | – | – | – | √ | – | – | – | – |
| *Russula sp 4* | – | – | – | – | – | – | √ | – | √ | – | √ | √ | √ | √ | √ |

(Continued)

| Species name | Above-ground | | | | | Root | | | | | Soil | | | | |
| --- | --- | --- | --- | --- | --- | --- | --- | --- | --- | --- | --- | --- | --- | --- | --- |
| | June | July | August | September | October | June | July | August | September | October | June | July | August | September | October |
| *Russula sp 5* | – | – | – | – | – | √ | √ | √ | √ | √ | √ | √ | √ | √ | √ |
| *Russula sp 6* | – | – | – | – | – | √ | √ | – | √ | √ | √ | √ | √ | √ | √ |
| *Russula sp 7* | – | – | – | – | – | √ | √ | √ | √ | √ | √ | √ | √ | √ | √ |
| *Russula sp 8* | – | – | – | – | – | – | – | – | – | – | – | – | √ | – | – |
| *Russula sp 9* | – | – | – | – | – | – | – | – | – | – | √ | – | – | – | – |
| *Russula sp 10* | – | – | – | – | – | – | – | – | – | √ | – | √ | √ | – | √ |
| *Russula sp 11* | – | – | – | – | – | – | – | – | √ | – | – | – | – | √ | √ |
| *Russula sp 12* | – | – | – | – | – | – | – | – | √ | – | – | √ | – | √ | – |
| *Russula sp 13* | – | – | – | – | – | – | – | – | √ | – | – | – | – | √ | – |
| *Russula sp 14* | – | – | – | – | – | – | – | – | – | – | √ | – | – | – | – |
| *Russula sp 15* | – | – | – | – | – | – | – | – | – | – | – | – | – | – | √ |
| *Russula sp 16* | – | – | – | – | – | – | – | – | √ | – | – | – | – | √ | – |
| *Russula sp 17* | – | √ | – | – | – | – | – | – | – | – | – | – | – | – | – |
| *Russula sp 18* | – | √ | √ | – | – | – | – | – | – | – | – | – | – | – | – |
| *Russula sp 19* | – | √ | – | – | – | – | – | – | – | – | – | – | – | – | – |
| *Russula sp 20* | – | – | √ | – | – | – | – | – | – | – | – | – | – | – | – |
| *Russula sp 21* | – | – | √ | – | – | – | – | – | – | – | – | – | – | – | – |
| *Russula sp 22* | – | – | √ | – | – | – | – | – | – | – | – | – | – | – | – |
| *Russula sp 23* | – | – | √ | – | – | – | – | – | – | – | – | – | – | – | – |
| *Russula sp 24* | – | √ | – | – | – | – | – | – | – | – | – | – | – | – | – |

of *Lactifluus* only appeared in late August, while there were no species in September and October (Fig. 2A).

## The composition changes on the growing seasonal basis of the Russulaceae fungal community from the roots

A total of 1,440 OTUs were detected from the root samples, 149 OTUs were unidentified fungi, and 1,291 OTUs were identified fungi, belonging to 4 phyla, 18 classes, 56 orders, 112 families, and 271 genera, of which 40 OTUs were identified to the Russulaceae, accounting for 2.77% of the entire fungal community, and they were clustered into 37 species, including 30 species of Russula, 6 species of Lactarius, and 1 species of Lactifluus (Table 1).

The Chao richness index of the Russulaceae in August had reached its peak, followed by September, and was the lowest in July, June and October. The Shannon diversity index in June, July and October indicated that their degree of diversity was all significantly higher than that in August and September, but there were no significant differences between August and September (Table 3).

The number of Russulaceae species in the root samples during the growing season, showed a trend of decreasing, then increasing, and then decreasing, and the variation in

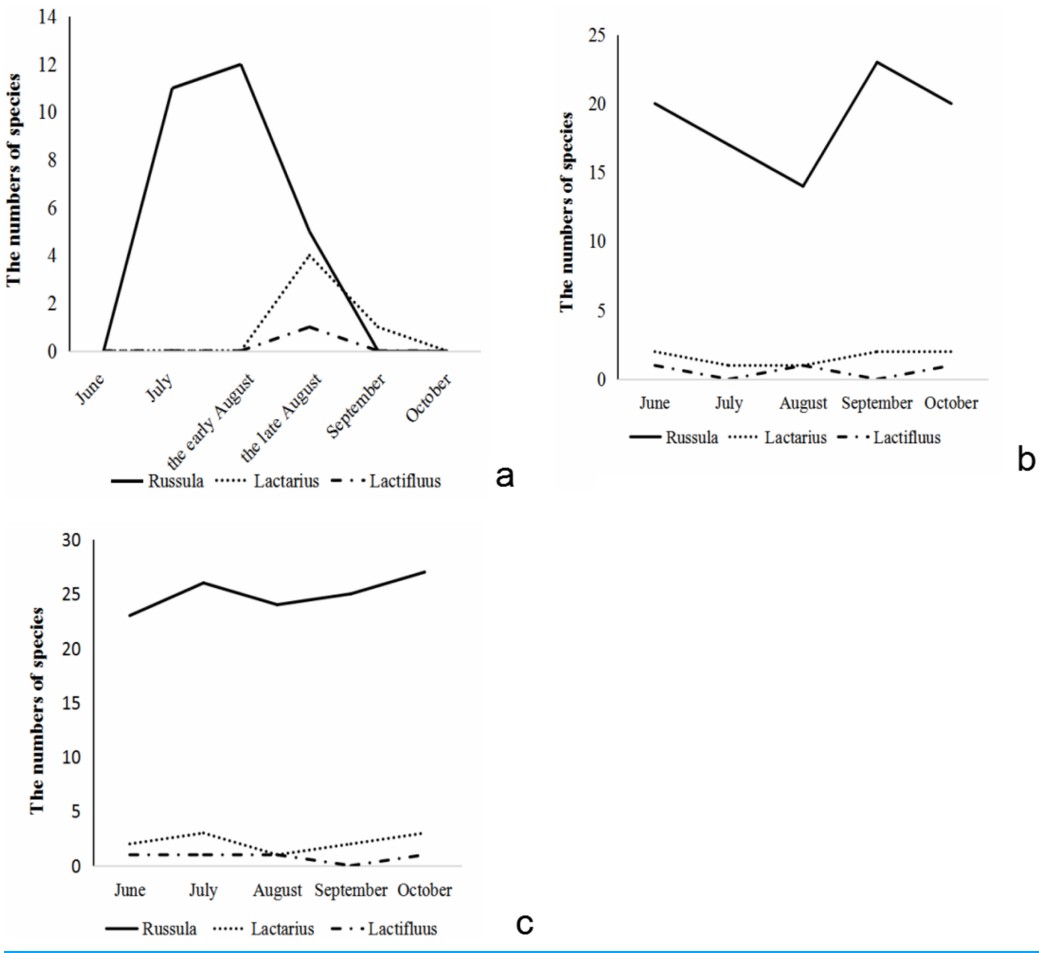

**Figure 2 The species number variation of *Russula* and *Lactarius* aboveground (A) in the root (B) and soil (C) from 5 months.**

**Table 2 Menhinick richness index, Shannon diversity index and Pielou's evenness index of the Russulaceae fungi on the aboveground associated with *Quercus mongolica* from June to October.**

|  | August | July | September | October | June |
|---|---|---|---|---|---|
| Menhinick[a] | 3.11 ± 0.00a | 2.04 ± 0.01ab | 1.00 ± 0.58b | 0.00 + 0.00b | 0.00 + 0.00b |
| Shannon[a] | 2.75 ± 0.01a | 2.44 ± 0.01b | 0.00 + 0.00c | 0.00 + 0.00c | 0.00 + 0.00c |
| Pielou's evenness[a] | 0.87 ± 0.01b | 0.92 ± 0.01a | 0.00 + 0.00c | 0.00 + 0.00c | 0.00 + 0.00c |

Note:
[a] $n = 5$. The values are the means ± standard errors. Different letters refer to significant differences between the months according to Tukey's test at $p < 0.05$.

the number of the species of *Russula* and *Lactarius* was basically the same. The number of species of *Russula* was the largest in September, followed by June, October and July, and the lowest in August. The number of species of *Lactarius* was the same and the largest in June, September and October, followed by July and August. The species of *Lactifluus* appeared only in June, August and September, and the number was the same (Fig. 2B).

**Table 3 Chao richness index and Shannon diversity index of the Russulaceae fungal communities in roots and soils from June to October.**

|  |  | June | July | August | September | October |
|---|---|---|---|---|---|---|
| Root | Shannon[a] | 2.24 ± 0.06a | 2.28 ± 0.06a | 1.47 ± 0.05b | 1.50 ± 0.04b | 2.22 ± 0.02a |
|  | Chao[b] | 26.69 ± 1.56 | 25.58 ± 4.77 | 39.57 ± 15.80 | 33.06 ± 4.65 | 26.11 ± 1.93 |
| Soil | Shannon[a] | 2.71 ± 0.02a | 2.18 ± 0.02b | 2.75 ± 0.02a | 2.12 ± 0.03b | 2.20 ± 0.02b |
|  | Chao[b] | 29.00 ± 0.00 | 45.64 ± 17.02 | 28.00 ± 0.00 | 29.00 ± 0.00 | 22.00 ± 19.05 |

Notes:
[a] $n$ = 5. The values are the means ± standard errors. Different letters refer to significant differences between months according to Tukey's test at $p < 0.05$.
[b] The values are the Chao mean ± Chao standard deviation (analytical) in each month.

## The composition changes on the growing seasonal basis of the Russulaceae fungal community from the soils

A total of 1,388 OTUs were detected from the soil samples, 205 OTUs were unidentified fungi, and 1,183 OTUs were identified fungi, belonging to 4 phyla, 18 classes, 52 orders, 106 families, and 245 genera, of which 46 OTUs were identified as members of the Russulaceae, comprising 3.31% of the entire fungal community, and they were clustered into 42 species, including 37 species of *Russula*, 4 species of *Lactarius*, and 1 species of *Lactifluus* (Table 1).

The Chao richness index of the Russulaceae showed that the richness was the largest in July, and the lowest richness in October, June was similar with August and September. The estimated Shannon diversity index indicated that the degree of diversity was significantly higher in June and August than that in July, September and October, but there were no significant differences between June and August, among July, September and October (Table 3).

The number of species of Russulaceae in the soil samples during the growing season first increased, decreased, and then increased, while the species number variation of *Russula* and *Lactarius* were basically the same. The number of *Russula* species was the largest in October, followed by July, September and August and June was the lowest. The number of *Lactarius* was the largest in July and October, followed by June and September, and the lowest in August. The species of *Lactifluus* appeared in all months except September, and the number of species was the same (Fig. 2C).

A comparison of the composition of the Russulaceae fungal community in the roots and soils indicated that there were 34 species of Russulaceae shared between the roots and soils; 3 species were unique in the roots, and 7 species were unique in the soils (Fig. 3). The Chao index ($p = 0.0740 > 0.05$) and Shannon index ($p = 0.5476 > 0.05$) of the Russulaceae between the roots and soils showed that there were no significant differences (Fig. 4). However, the abundances of some species differed significantly, the abundance of *R. cyanoxantha*, *R. pallidospora*, *R. foetens* and *R. azurea* was significantly lower in the roots than in the soils, while the abundance of *R. persicina* was significantly higher in the roots than in the soils (Fig. 5).
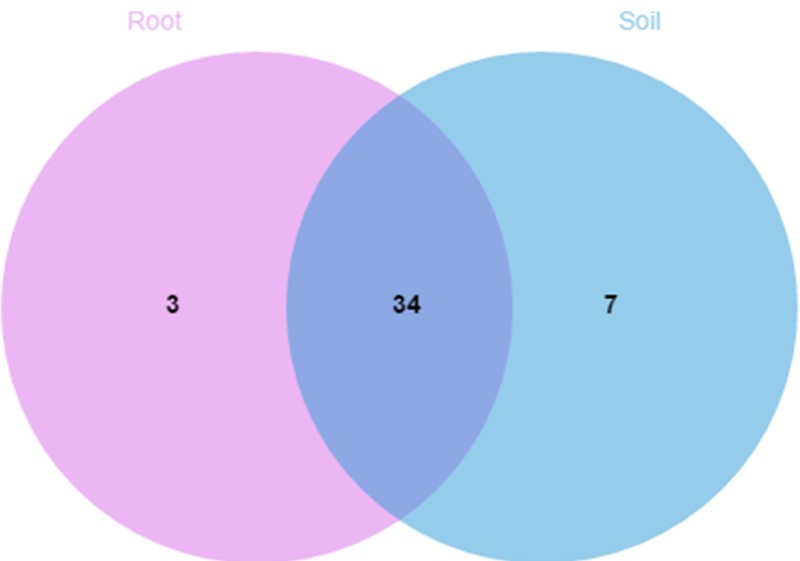

**Figure 3 Venn diagrams showing the number of Russulaceae species shared and unique to the root and soil.**

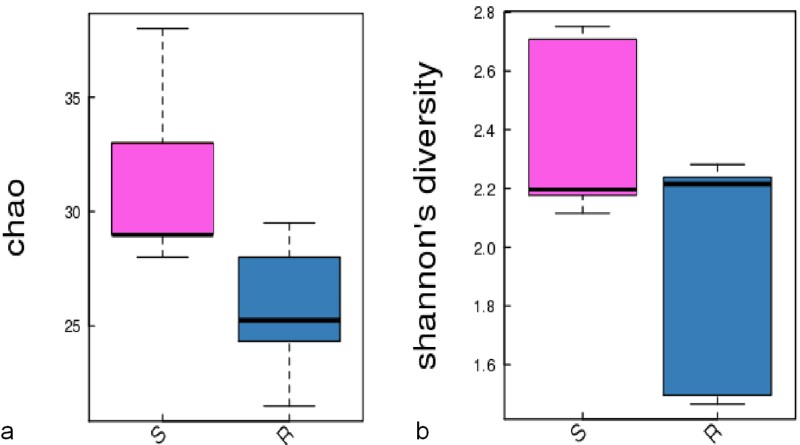

**Figure 4 The significant difference of the alpha diversity indices between the root group (R) and the soil group (S) with a confidence interval of 95%. (A) Chao diversity; (B) Shannon diversity.**

## Comparing the composition of the Russulaceae fungal community among the aboveground, roots and soils

There were 56 species of Russulaceae detected in the *Q. mongolica* forest, including 48 species of *Russula*, 7 species of *Lactarius*, and 1 species of *Lactifluus*, among which *Russula* was the dominant group (Table 1). The difference in the Shannon diversity index for the Russulaceae between the aboveground and roots and the aboveground and soils in July and August was analyzed by an ANOVA, respectively, which showed there were significant differences between the aboveground and roots ($p = 0.048 < 0.05$) and the aboveground and soil ($p = 0.001 < 0.05$) in July. In August, there was a significant difference between the

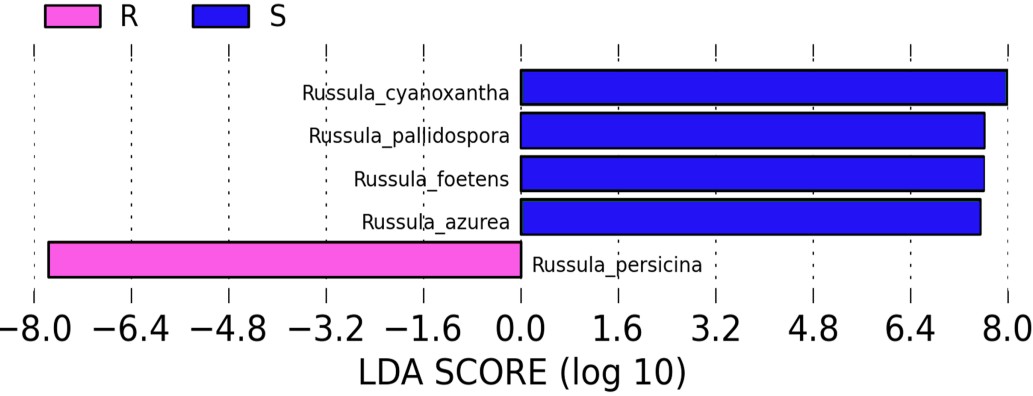

**Figure 5 LDA analysis of the root and soil groups. The default score of the LDA is 2.0, and the length of the bar chart represents the influence degree of the LDA score value and the species with significant differences between the different groups.** R indicates the root group, while S indicates the soil group.

**Table 4 The difference of Shannon index between the aboveground and root and between the aboveground and soil, respectively, in July and August.**

|  |  | Root | Soil |
|---|---|---|---|
| July | Sporocarps | 0.048 | 0.001 |
| August | Sporocarps | 0 | 1 |

Note:
Tukey's test at $p < 0.05$.

aboveground and roots ($p = 0.000 < 0.05$), but no significant difference with the soils ($p = 1.000 > 0.05$) (Table 4).

In June, there were no species on the aboveground; 18 species shared between the roots and soils; 5 species were unique in the roots; and 8 species were unique in the soils (Fig. 6A). In July, there were 2 species shared between the aboveground, roots and soils; 1 species only shared between the aboveground and soils; 14 species only shared between the roots and soils; 8 species were unique on the aboveground; 2 species were unique in the roots; and 12 species were unique in the soils (Fig. 6B). In August, there were 6 species shared between the aboveground, roots and soils; 1 species were only shared between the aboveground and roots; 9 were only shared between the species in the roots and soils; 11 species were unique on the aboveground; and 10 species were unique in the soils (Fig. 6C). In September, there was 1 species on the aboveground; 21 species were shared between the roots and soils; 3 species were unique in the roots, and 6 species were unique in the soils (Fig. 6D). In October, there were 22 species shared between the roots and soils; 1 species was unique in the roots, and 8 species were unique in the soils (Fig. 6E).

## Assessing the effects of abiotic factors on the Russulaceae fungal community composition

Varied results were revealed by the Pearson correlation between the alpha diversity index of the aboveground sporocarps in the *Q. mongolica* forest and nine abiotic factors.

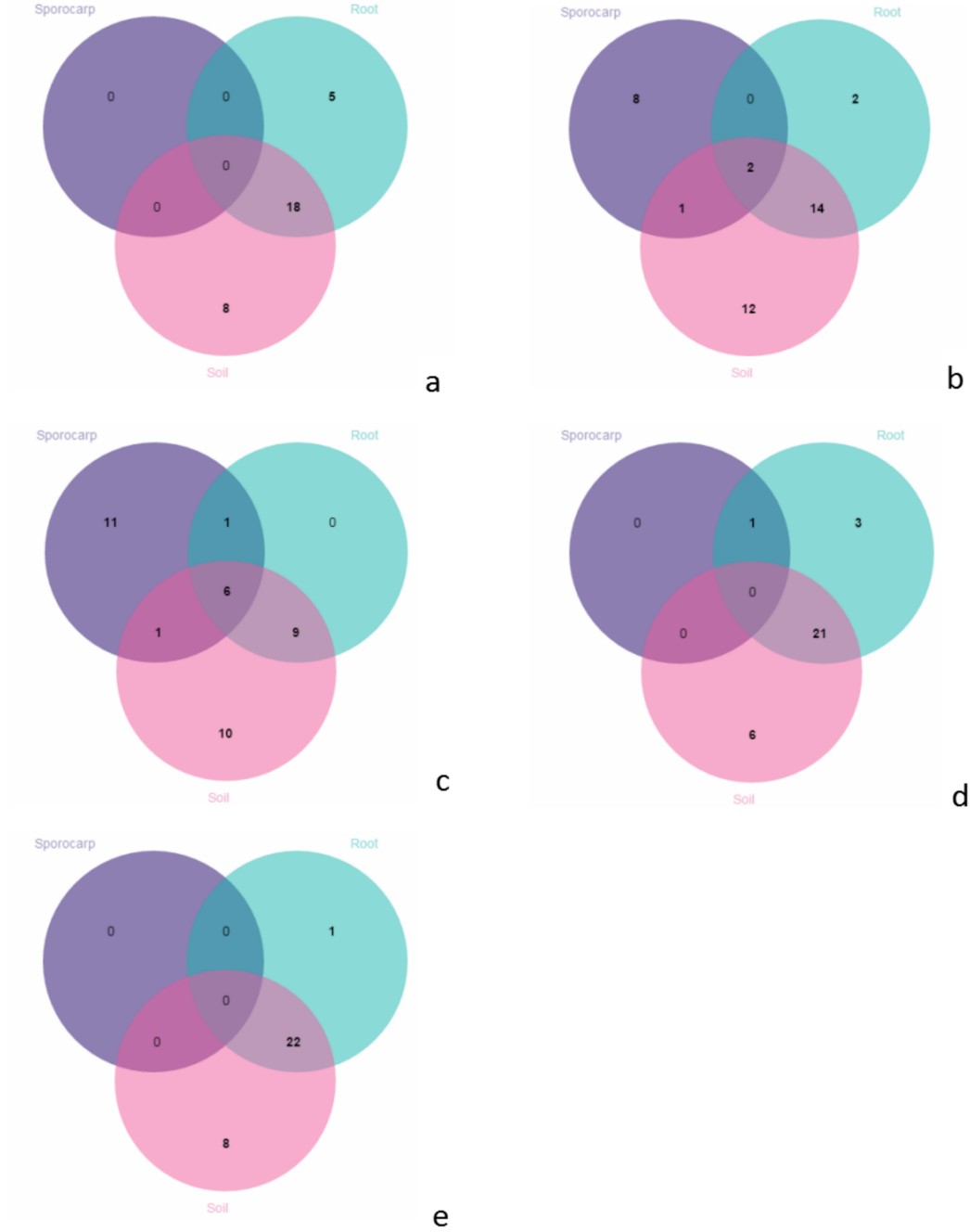

**Figure 6 Venn diagrams showing the number of species shared and unique between the aboveground, root and soil from the different months; (A) June; (B) July; (C) August; (D) September and (E) October.**

The Menhinick richness index showed a significantly negative correlation between the Temp ($r = -0.822$, $p = 0.045$) and RH ($r = -0.826$, $p = 0.043$), and a significantly positive correlation between the SOM ($r = 0.911$, $p = 0.012$) and SM ($r = 0.841$, $p = 0.036$). The Shannon diversity index had a significantly positive correlation with the MR ($r = 0.850$, $p = 0.032$) (Table 5). The RDA conducted for the Russulaceae fungal species

**Table 5 Correlation between the alpha diversity index of the Russulaceae fungi sporocarps on aboveground and environmental variables in a *Quercus mongolica* forest.**

|  |  | Temp | MR | RH | pH | N | K | SOM | SM | P |
|---|---|---|---|---|---|---|---|---|---|---|
| Menhinick | r | −0.822* | 0.176 | −0.826* | 0.773 | 0.109 | 0.705 | 0.911* | 0.841* | 0.194 |
|  | p | 0.045 | 0.739 | 0.043 | 0.071 | 0.838 | 0.118 | 0.012 | 0.036 | 0.712 |
| Shannon | r | 0.139 | 0.850* | 0.153 | −0.217 | 0.056 | −0.213 | 0.007 | −0.136 | 0.178 |
|  | p | 0.793 | 0.032 | 0.772 | 0.679 | 0.916 | 0.685 | 0.989 | 0.797 | 0.736 |
| Pielou | r | 0.753 | 0.613 | 0.781 | −0.753 | −0.104 | −0.756 | −0.78 | −0.788 | −0.339 |
|  | p | 0.084 | 0.196 | 0.067 | 0.084 | 0.845 | 0.082 | 0.067 | 0.063 | 0.511 |

**Note:**
  Significance level: $0.01 < ^{*} < 0.05$.

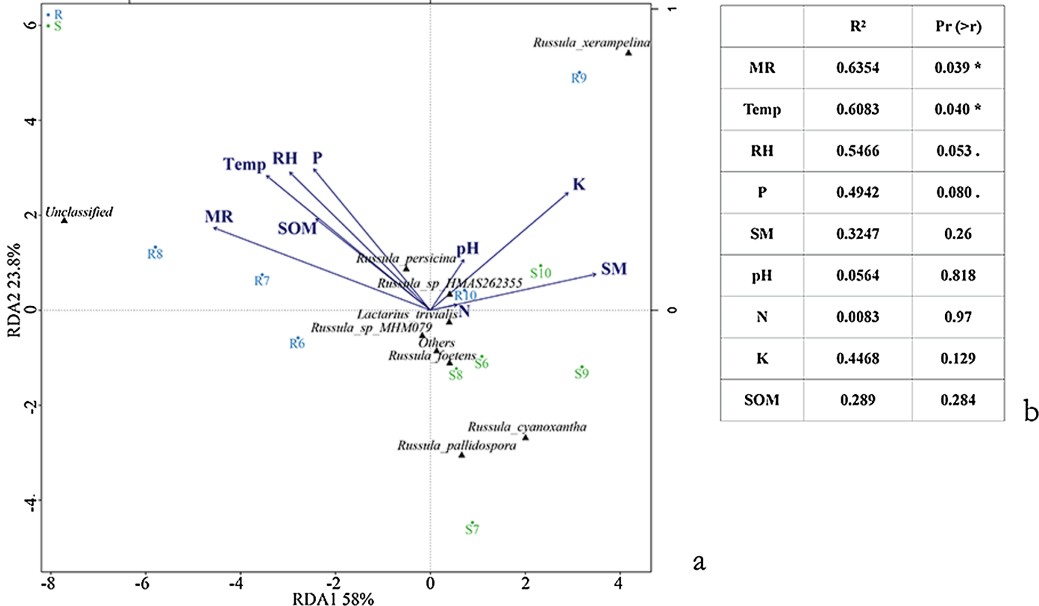

**Figure 7 Ordination diagram illustrating the effects of species (A) and (B) Temp, Rain, Air, SM, P, SOM, N, K and pH based upon an RDA analysis of the belowground Russulaceae fungal communities.** Significance codes: $0.1 > . > 0.05$; $0.05 > ^{*} > 0.01$.

belowground showed two significantly positive correlation variables: Temp ($r^2 = 0.6083$, $p = 0.040$) and MR ($r^2 = 0.6354$, $p = 0.039$). Other factors, including RH, SM, P, SOM, K, N and pH, had no significant influence on the distribution and diversity of the Russulaceae fungi in the roots and soils (Figs. 7A and 7B).

## Other fungal groups correlated with the composition of the Russulaceae fungal community

By performing a multivariate linear regression analysis in SPSS, we found that the relative abundance of three genera ($R = 1$, $p = 0.000$) in the root was linearly correlated with the relative abundance of *Russula*, and the linear regression equation was

**Table 6 Description of the functional groups on the genera significantly associated with the Russulaceae fungi from belowground.**

| | | Related genera | $R$ | $R^2$ | $P$ | Multiple linear regression equation | Guild type |
|---|---|---|---|---|---|---|---|
| Root | *Russula* | *Cortinarius* | 1 | 1 | 0 | Y1 = 2.058 + 0.153U1 − 0.450U2 + 4.609U3 | Ectomycorrhizal |
| | | *Exophiala* | | | | | Animal Pathogen or Undefined Saprotroph |
| | | *Herpotrichia* | | | | | Undefined Saprotroph |
| | *Lactarius* | *Inocybe* | 1 | 1 | 0 | Z1 = −0.014 + 0.169V1 + 0.008V2 − 0.003V3 | Ectomycorrhizal |
| | | *Nectria* | | | | | Animal Pathogen or Endophyte or Fungal Parasite or Lichen Parasite or Plant Pathogen or Wood Saprotroph |
| | | *Dictyochaeta* | | | | | Undefined Saprotroph |
| Soil | *Russula* | *Lachnum* | 1 | 1 | 0.001 | Y2 =−0.871 + 116.99W1 + 22.826W2 + 0.38W3 | Undefined Saprotroph |
| | | *Ilyonectria* | | | | | Undefined Saprotroph |
| | | *Cadophora* | | | | | Endophyte |
| | *Lactarius* | *Preussia* | 1 | 1 | 0.001 | Z2 = 0.145 + 0.901X1 − 0.003X2 + 0.084X3 | Dung Saprotroph or Plant Saprotroph |
| | | *Cortinarius* | | | | | Ectomycorrhizal |
| | | *Herpotrichia* | | | | | Undefined Saprotroph |

Note:
In the root, Y1, *Russula*; U1, *Cortinarius*; U2, *Exophiala*; U3, *Herpotrichia*; Z1, *Lactarius*; V1, *Inocybe*; V2, *Nectria*; V3, *Dictyochaeta*. In soil, Y2, *Russula*; W1, *Lachnum*; W2, *Ilyonectria*; W3, *Cadophora*; Z2, *Lactarius*; X1, *Preussia*; X2, *Cortinarius*; X3, *Herpotrichia*.

Y1 = 2.058 + 0.153U1 − 0.450U2 + 4.609U3 (Y1: *Russula*, U1: *Cortinarius*, U2: *Exophiala*, U3: *Herpotrichia*). The relative abundances of the three genera ($R = 1$, $p = 0.000$) were linearly correlated with *Lactarius*, and the equation was Z1 = −0.014 + 0.169 V1 + 0.008V2 − 0.003V3 (Z1: *Lactarius*, V1: *Inocybe*, V2: *Nectria*, V3: *Dictyochaeta*).

In the soil, three genera ($R = 1$, $p = 0.001$) correlated with *Russula*, and the equation was Y2 = −0.871 + 116.99W1 + 22.826W2 + 0.38W3 (Y2: *Russula*, W1: *Lachnum*, W2: *Ilyonectria*, W3: *Cadophora*). Three genera ($R = 1$, $p = 0.001$) correlated with *Lactarius*, of which the regression equation was Z2 = 0.145 + 0.901X1 − 0.003X2 + 0.084X3 (Z2: *Lactarius*, X1: *Preussia*, X2: *Cortinarius*, X3:*Herpotrichia*) (Table 6).

## DISCUSSION

### The composition changes on the growing seasonal basis of the Russulaceae fungal community on the aboveground, roots and soils

Sporocarps on the aboveground were identified using morphological and the Sanger sequencing method, and the species of Russulaceae in the roots and soils in a *Q. mongolica* forest were identified using the high-throughput sequencing method. Although the high-throughput sequencing removed the species of low relative abundance, more comprehensive data was still available. We were able to obtain a more detailed understanding about the community composition of Russualaceae than that provided by the traditional methods, which only rely on the morphological and molecular identification of the mycorrhizae belowground. To avoid the mismatch between the collected time and the ECM formation period of the ECM fungal species in the forest ecosystems, we collected samples twice per month during the growing season.

In this study, we found that the variation trend of the species number of the Russulaceae aboveground was consistent with our previous hypothesis, the period of production of

the sporocarps on the aboveground was in July and August. The species number of *Russula* in a *Castanopsis cuspidata* (Thunb.) Schottky forest in Japan formed a large peak on July 24, and the number was the lowest after mid-August (*Murakami, 1987*). *R. nigricans* in Korea was commonly found during the summer and fall (*Park et al., 2014*). While the variation trend of the species number of Russulaceae in the roots was not consistent with our previous hypothesis, the species number was the highest in September. The richness of the ECM fungal species in the root of *Q. ilex* was the highest in the fall (September–November) (*Román & Miguel, 2005*), which was similar to our results. Most of the ECM species (*Russula*, 1%; *Lactarius*, 0.3%; and *Amanita*, 0.05%) in the soil of a *Q. petraea* (Matt.) Liebl. forest in the Czech Republic had a lower relative abundance in the spring, which was higher in the summer (*Russula*, 7%; *Lactarius*, 6%; and *Amanita*, 3.4%) (*Voříšková et al., 2014*; *Castaño et al., 2017*). The diversity and richness of the Russulaceae species in the soil of the *Q. mongolica* forest in our study was higher in the summer than in the autumn.

Some species only presented aboveground and were absent in the roots and soils in the *Q. mongolica* forest, such as *R. atroglauca*, *R. aurata*, *R. delica* and *L. evosmus*, among others, which are ECM fungi. The reason that these species were absent from the root and soil may be that their trimmed reads had less than 75% of their original length in the data processing and resulted in their removal. It could also be that the ECM fungi aboveground could transport their spores over long distances to other places through the wind, resulting in species migration (*Roy et al., 2008*; *Hirose, Shirouzu & Tokumasu, 2010*; *Vellend, 2010*; *Vincenot et al., 2012*; *Sheedy et al., 2015*; *Boeraeve, Honnay & Jacquemyn, 2018*; *Koizumi, Hattori & Nara, 2018*). Thus, we could not exclude this possibility. Other species, such as *L. pyrogalus*, *R. cyanoxantha*, and so on, only presented in the roots and soils, no sporocarps formed on the aboveground, which may be owing to environmental conditions that were not suitable for the formation of the sporocarps.

## Assessing the effects of abiotic factors on the Russulaceae fungal community composition

The composition of the Russulaceae fungal community in the *Q. mongolica* was driven by the climate and soil. The composition of Russulaceae aboveground was significantly affected by the Temp, MR, RH, SM and SOM, which were consistent with our hypothesis that Temp and MR were the important factors affecting the composition and distribution of the Russulaceae fungi aboveground. The abundance of ECM fungi in a Mediterranean forest during the same month was dependent primarily on the temperature, high temperatures limited the growth of the ECM fungi at the beginning of the fruiting season but tended to enhance it towards the end (*Karavani et al., 2018*). Rainfall affected the abundance of ECM fungi by affecting SM; therefore, suitable rainfall and SM can promote the abundance of the ECM fungi (*Salerni et al., 2002*; *Ogaya & Peñuelas, 2005*). In addition, the composition of the Russulaceae fungal community in the roots and soils was also significantly affected by Temp and MR, which were consistent with our hypothesis. Warming increased the abundance of the ECM fungi in the roots of pine forests (*Saravesi et al., 2019*) and increased the mycelial biomass of *L. vinosus* (Quél.)

**Table 7 Months of the abiotic factors associated with the Russulaceae fungi.**

| Month | MR (mm) | RH (%) | Temp (°C) | SM (%) | pH | N (mg·kg⁻¹) | K (mg·kg⁻¹) | SOM (%) | P (mg·kg⁻¹) |
|---|---|---|---|---|---|---|---|---|---|
| July | 251.60 | 87.00 | 20.80 | 57.08 | 6.05 | 321.67 | 523.10 | 15.69 | 34.90 |
| August | 145.40 | 86.00 | 17.30 | 61.61 | 6.14 | 321.35 | 610.58 | 17.67 | 32.32 |

Bataille in soil at the Natural Park of Poblet (*Castaño et al., 2017*). In addition, enough rainfall can promote the growth of ECM fungi in the root of the dry dipterocarp forest (*Disyatat et al., 2016*).

Furthermore, from our survey, we also roughly summarized the range of nine abiotic factors, which were suitable for the growth of the Russulaceae fungal sporocarps. The range of nine abiotic factors in the fruiting period from July to August is found in Table 7 and includes the MR (145.40 mm–251.60 mm), RH (86.00–87.00%), Temp (17.30–20.80 °C), pH (6.05–6.14), SM (57.08–61.61%), pH (6.13–6.14), N (321.35 mg·kg⁻¹–321.67 mg·kg⁻¹), K (523.10 mg·kg⁻¹– 610.58 mg·kg⁻¹), SOM (15.69–17.67%) and P (32.32 mg·kg⁻¹–34.90 mg·kg⁻¹), respectively (Table 7). We hope that these results will provide data to enhance research on the cultivation of mycorrhizal mushrooms.

## Correlations of other functional fungi that play an important role in the roots and soils with Russulaceae fungal community composition

We used a multiple regression analysis to establish the multiple regression analytical models to analyze the influence of other fungal species on the Russulaceae fungal community composition, which could negate the influence of a third genus when the relationship of two genera with each other was measured. Moreover, we obtained the most suitable equation model by comparing the goodness of fit, correlation coefficient and significance level, which showed that the multiple regression analysis model was more appropriate.

The relative abundance of *Russula* in the roots was linearly correlated with the relative abundance of *Cortinarius*, which co-appeared with *Russula* in many different forest roots (*Jang & Kim, 2012*; *LeDuc et al., 2013*). The relative abundance of *Russula* in the soils was linearly correlated with the relative abundance of *Lachnum*, which also appeared in soil of a *Lithocarpus densiflorus* forest in northern California (*Bergemann & Garbelotto, 2006*). However, there is no evidence to show their relationship, and it is not clear in what manner these genera affect the Russulaceae fungi and merits further study.

## CONCLUSIONS

This study revealed the composition variation of the Russulaceae fungal community aboveground and in the roots and soils during the growing season of a *Q. mongolica* forest. The composition of the Russulaceae fungi had been significantly affected by the average month air temperature, monthly rainfall, average month relative humidity, soil organic matter and soil moisture. This will provide a scientific basis for the further cultivation of the Russulaceae fungi. However, it is not clear in what manner some other genera affect the Russulaceae fungi, and this possibility merits further study.

### Funding

This research was supported by the National Natural Science Foundation of China (No. 31600020) and the Overseas Expertise Introduction Project for Discipline Innovation (111 center) (No. D17014). The work was primarily performed in the Engineering Research Center of Edible and Medicinal Fungi in Jilin Agricultural University for Jilin Province. The funders had no role in study design, data collection and analysis, decision to publish, or preparation of the manuscript.

### Grant Disclosures

The following grant information was disclosed by the authors:
National Natural Science Foundation of China: 31600020.
Overseas Expertise Introduction Project for Discipline Innovation (111 center): D17014.
Engineering Research Center of Edible and Medicinal Fungi in Jilin Agricultural University for Jilin Province.

### Competing Interests

The authors declare that they have no competing interests.

### Author Contributions

- Pengjie Xing conceived and designed the experiments, performed the experiments, analyzed the data, prepared figures and/or tables, authored or reviewed drafts of the paper, and approved the final draft.
- Yang Xu conceived and designed the experiments, performed the experiments, analyzed the data, prepared figures and/or tables, authored or reviewed drafts of the paper, and approved the final draft.
- Tingting Gao analyzed the data, prepared figures and/or tables, and approved the final draft.
- Guanlin Li performed the experiments, analyzed the data, authored or reviewed drafts of the paper, and approved the final draft.
- Jijiang Zhou performed the experiments, analyzed the data, prepared figures and/or tables, and approved the final draft.
- Mengle Xie conceived and designed the experiments, prepared figures and/or tables, authored or reviewed drafts of the paper, and approved the final draft.
- Ruiqing Ji conceived and designed the experiments, authored or reviewed drafts of the paper, and approved the final draft.

### Field Study Permissions

The following information was supplied relating to field study approvals (i.e., approving body and any reference numbers):

Field experiments were approved by the Wudalianchi Scenic Area Nature Reserve, Heilongjiang Province, China for field work in the Jiaodebu forest farm construction area.

## Data Availability

The raw measurements are available in the Supplemental Files.

## Supplemental Information

Supplemental information for this article can be found online at http://dx.doi.org/10.7717/peerj.8527#supplemental-information.

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
