# Peer review of "The community composition variation of Russulaceae associated with the Quercus mongolica forest during the growing season at Wudalianchi City, China"

_PeerJ, doi:10.7717/peerj.8527_

## Round 0.1 · original submission · Minor Revisions

Please, when submitting the revised version of the paper, provide rebuttals to the comments of the reviewers. In particular, consider carefully the criticism raised by Rev2, who noted that "the largest apparent weakness in this study’s approach is the scope of only looking at a single site over a single fruiting season. Without a multi-year replication of the study or comparison with another site with different soil characteristics or climate, it is difficult to infer whether these correlations with changing pH or temperature is actually impacting community composition."

This requires a deeper discussion of the results and description of the limitation of the study.

·

Basic reporting

No comment

Experimental design

Original primari research within Aims and Scope of the journal.
Methods described with sufficient detail and information to replicate.

Validity of the findings

All underlying data have been provided; they are robust, statistically sound, & controlled.

Conclusions are well stated, linked to original research question & limited to supporting results.

Additional comments

This study aims at understanding the change in the composition of the Russulaceae fungal community aboveground, in the root and soil during the growing season in a Quercus mongolica forest in Wudalianchi City, China. This study provides an important scientific basis for the further cultivation of the fungi of Russulaceae family.

Here my following minor comments:

Line 41: Replace “ectomycorrhizal” with “mycorrhizal”.
Land plant species can form mutualistic symbiosis with endo- and ectomycorrhizal fungi. Review the whole period (from line 41 to line 45).
Line 51: Author(s) of species name must be provided when the scientific name of any plant or fungal species is first mentioned. Correct through the whole manuscript and Tab. 1. For plant see “The Plant List (http://www.theplantlist.org)”; for fungi see “index fungorum (http://www.indexfungorum.org/names/Names.asp)”.
Lines 96-97: (1) What the…. (2) How abiotic…..(write all lowercase)

Lines 99 and 103: (1) During the… (3) Fungi sporocarp…..(write all lowercase)

Line 109: 48°37′0 W 126°11′0 N, Latitude and longitude are reversed. Correct as follows: Lat 48.616667° Long 126.183333°. 303m??? Is it perhaps 300 m asl?

Reviewer 2 ·

Basic reporting

The English language is grammatically fine; however, the clarity of the language is sometimes lacking. An example is line 49 – the current phrasing does not make very much sense.

Title – please define what the growing season is. Is this an assumed fungal growing season or the plants

Line 40- I believe you mean mycorrhizal fungi, not ectomycorrhizal fungi. Only 2% of plants associate with ECM.

Materials and Methods

There is no explicit statement for where the data will be deposited.

Line 113- How far apart were these plots? This is important for interpreting results from Figure 7. Also, what is the plant composition of these forests. Only Quercus?

Line 159- Is there a citation for the “potassium dichromate volumetric method” and other soil biogeochemistry assay approaches?

Line 167- I’m unfamiliar with the term “tags” in amplicon sequence data. Could you explain more?

Line 229- I am surprised that September and October were so sparse. Would we normally expect fruiting at this time (I assume the time period correlates with the fruiting season)?

Line 245- Table is misspelled here.

Line 330- It is still unclear how morphology and sequencing were used to identify sporocarp species. GenBank is an unsuitable reference database for identifying species of Russula in most cases.

Line 336- mycorrhizae or sporocarps or both?

Line 342- Comparisons of phenology between different geographic locations based on month is irrelevant unless it can be demonstrated that the seasonality is similar to your own study site

Line 357- This argument does not make much sense. More likely these species have a very patchy distribution across the root landscape and do not widely explore soil with extramatrical hyphae. See Agerer (2001).

Line 364- How can you say climate impacts the community when you have one study site? Also, the soil does not vary across months…

Taxonomic entities should be italicized in References

Figure 5- Why not have Root and Soil in the legend?

Experimental design

The sampling and statistical approaches are sound. Some of the methodologies should be cited in References for "standard approaches".

Validity of the findings

The study reveals some interesting phenological patterns of above/below-ground diversity of Russulaceae. It seems odd to compare soil characteristics over the span of only 4 months. I’m not convinced that differences recovered within the same forest in soil characteristics are relevant

Additional comments

Xing & Xu et al. present a well-executed study of phenological patterns of Russulaceae in a Quercus dominated forest during the growing season. The study is multifaceted through the use of sporocarp sampling and metagenomic amplicon sequencing to characterize the entire Russulaceae community. Additionally, a suite of biogeochemical analyses and environmental variables were used to compare correlations for community composition. This is an interesting and novel approach.

1. The largest apparent weakness in this study’s approach is the scope of only looking at a single site over a single fruiting season. Without a multi-year replication of the study or a comparison with another site with different soil characteristics or climate, it is difficult to infer whether these correlations with changing pH or temperature is actually impacting community composition. The study demonstrates the patchiness of Russulaceae distribution, even within the same forest.
2. I am a little concerned about the transparency of species determination in this study. The authors should explain how in greater detail how morphology and the “Sanger method” was used to identify species. Was there an identity cutoff used for Sanger sequences?

---

## Round 0.2 · accepted · Accept

The new version of your article has satisfactorily incorporated previous reviewers' requests. The work is now suitable for publication. Thanks for considering PeerJ and best wishes for your future research

·

Basic reporting

Professional article structure, figures, tables.

Experimental design

Method described with sufficient detail

Validity of the findings

Conclusions are well stated, linked to original research question & limited to supporting results.

Additional comments

I have no corrections.